# Forensic Geology Applied to Decipher the Landslide Dam Collapse and Outburst Flood of the Santa Cruz River (12 November 2005), San Juan, Argentina

Juan Pablo Milana [1,*] and Philipp Geisler [2]

1    Instituto de Geologia, CONICET, Universidad Nacional de San Juan, Av. Ignacio de la Roza y Meglioli, San Juan 5401, Argentina
2    Departamento de Física, Universidad de Atacama, Av. Copayapu 485, Copiapo 1530000, Chile; philipp.geisler@uda.cl
*    Correspondence: jpmilana@gmail.com or milana@unsj-cuim.edu.ar

**Abstract:** A well-known landslide dam that collapsed and generated a large outburst flood is used to show the importance of forensic geology analysis, which is the on-site multidisciplinary study of geohazards carries out as soon as possible after their occurrence; this study is focused on understanding the complete spectrum of all mechanisms that caused the disaster. Diagnostic elements of all natural processes fade with time, allowing for progressively divergent interpretations that may impact the appropriateness of potential mitigation actions, as we demonstrate. The multidisciplinary field control of the abrupt rupture of a natural dam on the Santa Cruz River on 12 November 2005, that released c. 37 million m$^3$ of water and sediment, can radically change the interpretation of how this dam collapsed. In situ sedimentological, geomorphological and topographical analyses of the remains of the collapsed natural dam suggest it was built in two mass-wasting episodes instead of one, as previously interpreted, involving different slide materials. The first episode matches previous interpretations; a landslide that evolved into a rock avalanche, generating an initial dam of high stability due to its density, and observed angles of repose. This dam was not removed completely during the rupture, but rather suffered minor erosion at its top by the flood drag effect. The second episode is interpreted as a snow-dominated mixed avalanche, reaching much greater heights on the opposite side of the valley. This avalanche is estimated to be 85% snow, 8% debris and 7% ice-cemented permafrost fragments, and is evidenced by a thin residual deposit draping the valley sides, as most of this deposit melted out before any field control was undertaken. The growth of the lake level, along with the dam weight loss due to ablation, generated the hydrostatic instability that caused the floating of the central sector of this second dam and the violent evacuation of the water, similar to a jökulhlaup. This analysis explains the partial dam collapse, sudden water release and the preserved field evidence. This different interpretation suggests that the mitigation actions already taken can be improved and that monitoring systems are urgently needed. A rapid and professional assessment of any large-scale geohazard site would be the way to avoid interpretation discrepancies, and to guarantee that mitigation actions taken are adequate. Learning from this event may help decision makers to take better mitigation measures and potentially save lives.

**Keywords:** jökulhlaup; natural dam; landslide; avalanche; governance

## 1. Introduction

### 1.1. The Forensic Geology Concept

In this contribution, we use the concept of forensic geology to better understand a geological process where the interpretations involve a certain level of debate. Ruffel et al. [1], based on the publications of Murray et al. [2–4] and Pye et al. [5,6], defined forensic geology as "the use of geological methods and materials in the analysis of samples and places that maybe connected with criminal behaviour or disasters. Geology in this sense encompasses

geological methods of analysis (geophysics, petrography, geochemistry, microscopy, micropalaeontology)". In the practice, forensic geology has been traditionally used to resolve crime scenes and legal issues, and a minor application to disasters has been observed. It has been mentioned that "dams formed from landslides, glacial ice, and late-neoglacial moraines present the greatest threat to people and property, of numerous kinds of dams that form by natural processes" [7]; thus, we believe that the definition applies to the matter at hand. "Forensic" is an adjective derived from the Latin concept of "forum", which is the physical or virtual place were arguments were publicly debated. In fact, the first definition of forensic as an adjective, according to the Merriam-Webster Dictionary, is "belonging to, used in, or suitable to courts of judicature or to public discussion and debate". On the other hand, its meaning as a noun is "an argumentative exercise". Therefore, both the noun and the adjective perfectly apply to the job which a professional performs when a hazardous natural event has occurred and a proper analysis of the remaining evidence needs to be undertaken in order to (a) explain the chain of events that caused the natural disaster, and (b) suggest measures to avoid future losses in the case the process occurs again.

In this paper, we demonstrate the need of forensic geological skills to reach the two abovementioned goals, using the largest single flood recorded in the last half century in the Central Andes as an example. As shown by the following analysis, the events that lead to this large flood are susceptible to multiple and very different hypotheses with potential remediation actions that could be quite opposing. The importance of an open discussion of these events in professional forums is crucial for remediation and prevention of similar disasters. The role of the professionals involved in defining the chain of events before, during and after the disaster is fundamental, and the objective of this contribution is to demonstrate the need for different disciplines to solve the enigma posed by a complex geohazard.

We propose the Santa Cruz outburst flood as an exercise in forensic geology, because we identified potential inconsistencies between published interpretations and different reports, particularly verbal reports of witnesses that did not reach the professional media. All previous studies reached the same conclusions: a landslide turned into a rock avalanche, creating a natural dam, in January 2005, the failure of which released a large flood on 12 November 2005 [8–16]. The published explanations of the failure include overtopping by increasing lake levels [8,10–13], overtopping by tsunami [9,10], slope instability [10], seismic shaking [10] and in one case the dam may have collapsed when hit by a rock mass [14]. This particular natural dam blocked about 650 m of the stream reach (Figures 1 and 2) and its overspill point was 71 m high [11], suggesting its slope facing down river was 6–7°. Debris-made dams are formed every year in different courses of this arid segment of the Andes, with comparable repose angles, but they are usually progressively backwards eroded by dam overflow and downcut without an abrupt collapse or sudden release of dammed water. Thus, this unique event is a clear anomaly, and requires a more detailed analysis.

On the other hand, natural dam failure is very common. According to Costa and Schuster [7], 27% of landslide dams investigated fail within one day and another 50% fail within ten days, and they identify overtopping as the most common cause of failure. The Erizos Lake dam took more than 10 months to collapse. According to the revision of Zheng et al. [17], the proportion of failed landslide dams with width/height ratios below 15 is 87%, but this reduces to 65% for landslide dams with ratios larger 30. The Erizos Lake dam ratio was 9.5, so it was failure prone. However, this unique outburst flood constituted a stand-alone anomaly and required, in our view, detailed on-site field control. The lake size created by the dam also represented an anomaly for this region, motivating the question: How did this dam-break really arise?

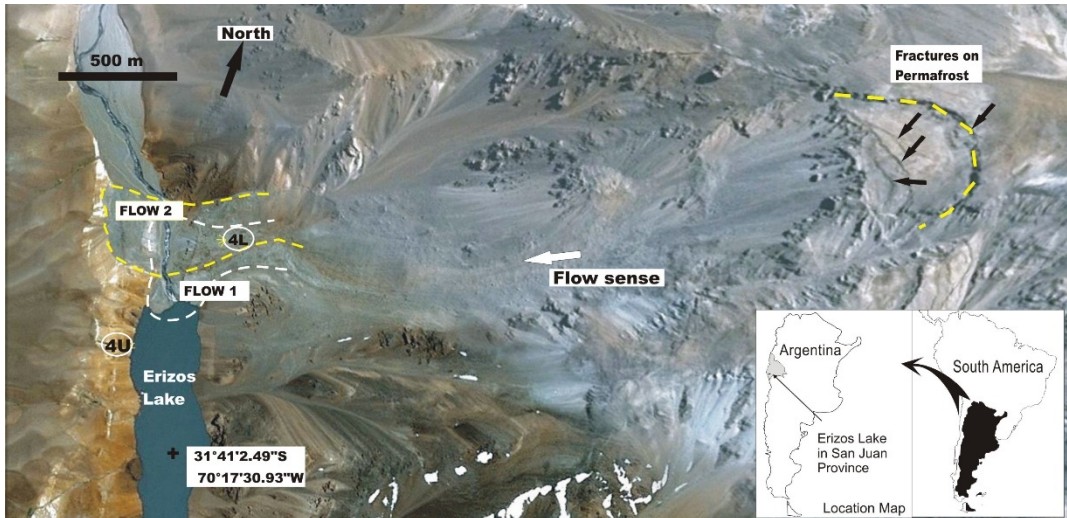

**Figure 1.** Satellite image of the landslide location, avalanche runway and mixed flows forming the dam complex. It shows the source area with a potential reservoir of permafrost, rock and debris limited by tensional cracks (grey arrows) that define a large block that already descended 35 m. The extent of Flow 1 and 2 deposit remains observed in the field is indicated. The cross marks the point of the coordinates. Viewpoints of ground photomosaics are indicated as 4U and 4L (see below).

The present analysis is based on the "remnant body" of the deposit, and this is why we use the term "forensic geology", as the technique to solve the enigma of the Santa Cruz River flood. As evidence may become more misleading with time passed since the event, the precept that a natural disaster site should be inspected as soon as possible after the event is widely known. This was not the case and thus, all reconstructions are arguable. Our reconstruction, several years later, suggested that a snow avalanche played a crucial role; however, it is clear that none of that snow survived a long time after the dam failure. Thus, interpretations would change according to the remaining evidence and the background of the observer, indicating the importance of a multidisciplinary professional field inspection as soon as possible after a collapse.

Aligned with the above reasoning, the objective of this contribution is to show that a well-known and widely published natural disaster can be suited to different interpretations, beyond logical doubts, using different evidence to sustain those different interpretations. A secondary objective is to give more value to detailed on-site field studies. Finally, we want to drive attention to the need of better governance in cases of natural hazards. An appropriate governance on this matter would lead to (a) an inspection as soon as possible by experienced professionals with knowledge of the involved natural processes; (b) the deployment of appropriate mitigation actions to protect the endangered population; and (c) the identification of potential similar dangers and near-real-time monitoring of potentially hazardous processes. This topic will be discussed later using this case study.

**Table 1.** Altitudes defined from the detailed topographic survey (Figure 2).

| Point on Figure 2 | Feature Mentioned in the Text | Altitude m.a.s.l. |
| :---: | --- | :---: |
| 1 | Valley floor before closure at the dam centre | 2940 |
| 2 | Highest flood terrace at canyon exit | 2950 |
| 3 | Lowest flood terrace at canyon exit | 2944 |
| 4 | Observed closure of Flow 1 | 2968 |
| 5 | Lake level intermediate mark | 2972 |
| 6 | Observed closure of Flow 2 | 3009 |
| 7 | Lake level highest mark | 3007 |
| 8 | Lake level after flood | 2957 |
| 9 | Lake level today | 2949 |

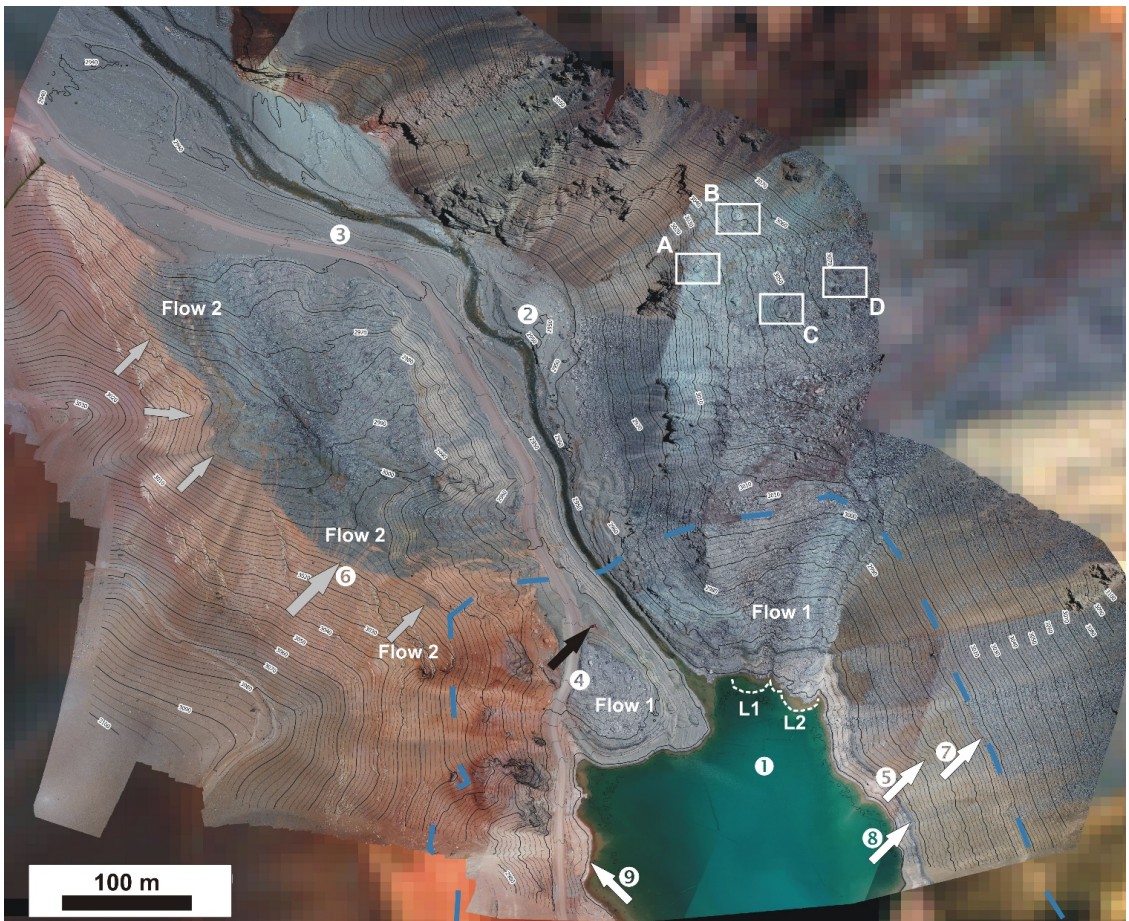

**Figure 2.** High-resolution topographic survey (6.94 cm GSD) processed with Pix4D to analyse the altimetric relationships of different elements related to the natural dam collapse. (A–D) Rectangles indicating higher resolution captions details of permafrost boulders relics (see below) White arrows indicate the position of the highest intermediate and the water level mark developed over 15 years after the flood draining event. The additional 8 m-level drop is due to road-building actions. Black arrow points to a known reference (truck is 5.5 m long). Grey arrows indicate different lobes of Flow 2, the largest one shows the highest topographic reach of Flow 2 deposits. L1 and L2 are Flow 1 lobes that clearly extend into the lake depth. The photomosaic made by 340 drone images is draped over the image showing the largest attained area of the Erizos Lake, marked by the hatched blue line. Encircled numbers identify the positions of features mentioned in Table 1.

### 1.2. The Santa Cruz River Jökulhlaup

The geology, geomorphology and previous studies related to the damming of the Santa Cruz River in 2005 have been duly presented in various works published after the event [8–11]. For this reason, in situ remains of the natural dam which blocked the Santa Cruz River, along with elements related to the dam collapse and resulting flood, are analysed here. We use jökulhlaup to describe this flood (an Icelandic word introduced by Thorarinsson [18]), rather than GLOF (glacier lake outburst flood), based on the interpretations discussed later. As previous works supported few or no field observations from the dam site, this contribution focuses on field observations and their further analysis. Field data may radically change existing interpretations.

Dam deposits allow the interpretation of the final transport process before sediment comes to rest, but can infrequently suggest the initial processes of movement [7], since transport processes of most mass-wasting events undergo changes as they move along its path. Moreover, slide materials may change due to deposition of lagged materials or inclusion of material on the run. Thus, the analysis of the areas of the initial rupture, the

evidence left along the pathway of the moving mass, and the final deposit characteristics point to different sedimentary processes: erosion, transport and deposition. Erosion and transport were satisfactorily analysed in previously published works; although, they did not mention the presence of melted permafrost fragments along the pathway. This is the reason this analysis is focused on the depositional remains of the mass-wasting event that created the natural dam, in order to find clues to identify in more detail the reasons for this unique outburst flood.

Due to the remoteness, the first field control of this site by one of authors (J.P.M.) took place in May 2012, after the mining company controlling the access road granted permission. The identification of two superimposed flow deposits of different types created the need for a detailed topography of the dam complex to give proper levels to the different elements observed. Two photogrammetric UAV surveys were carried in 2021, that allowed the three-dimensional aspects of each deposit identified to be defined, and the relation between the two potential dam heights and two strong watermarks generated during the Erizos Lake growth to be confirmed.

The satellite imagery shows that, late on 8 January 2005, a landslide blocked the Santa Cruz River and a lake started to form. Satellite imagery of 15 January 2005 shown by D'Odorico et al. [8] indicates that the dam deposit spread along the Santa Cruz River valley for c. 650 m of its course. The dam can be classified as a Type II dam [7], which are larger, span the entire valley floor, in some cases deposit material uphill on the opposite valley side and are related to the formation of bigger and more dangerous lakes. Hermanns et al. [19] extended the classification, where this dam is categorized as Type II e, due to the up- and down-valley spread. Subsequent satellite images do not show any evidence of another mass flow. However, a two-step areal growth of the lake occurred until the flood partially emptied it on 12 November 2005, as shown in Figure 3. According to D'Odorico et al. [11], c. 32 million cubic meters of water were suddenly released along the drainage network, affecting c. 200 km of the course of the river downstream. The flood filled two large dams within few hours, until both overspilled. Fortunately, no casualties were recorded, but much material loss and damage occurred along several civil constructions along the San Juan River system [8–11].

The dam collapse hypothesis proposed here is radically different to previous ones. We confirm the initial dam-building rock avalanche. However, our field observations suggest a second flow occurred, which added volume to the dam, rising its closure point by c. 35 m vertically (Table 1, Figure 4). Given the 3D scenario, we interpret that the second flow was volumetrically larger and mostly composed of snow/firn. A recreation of them, together with a 3D view of the main three lake levels, is given by Figure 5. Both flows also carried many large permafrost fragments, and are referred to as Flow 1 and Flow 2 hereafter.

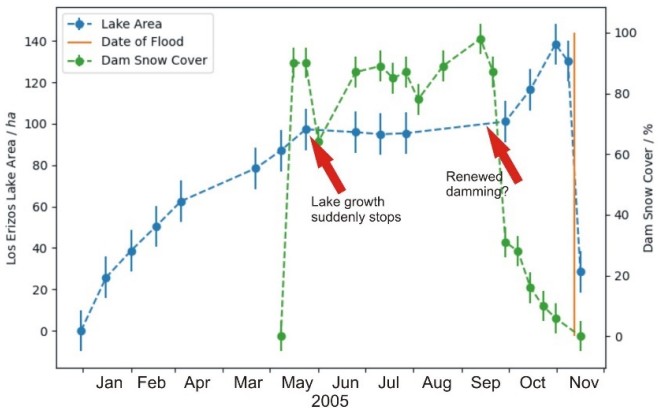

**Figure 3.** Los Erizos Lake area development showing the two-step lake growth, according to LANDSAT satellite images with visible lake perimeter. The percentage of snow cover over the dam complex is added (c. 20 Has), showing the coincidence of the maximum snow coverage and the second stage of lake growth.

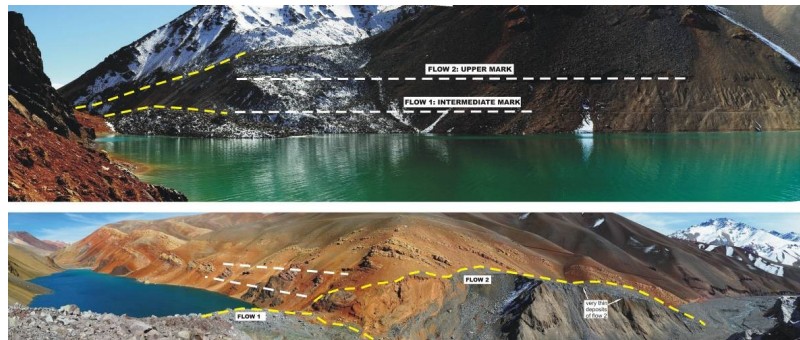

**Figure 4.** Two ground-acquired photomosaics showing the dissected dam and the interpreted flows from different perspectives. Viewpoints are indicated as 4U for the upper photomosaic and 4L for the lower one in Figure 1. Both mosaics show the two most important water level marks (white dashed lines) and the top of the two different flow deposits (yellow dashed lines). Upper mosaic: Shows the match between major lake-level water marks and the two deposits recognized in the field survey. Compare this mosaic with Figure 2. Lower mosaic: Taken from the avalanche couloir to the distal end of the flows. Arrows point to the opposite wall of the pre-existing canyon wall, reworked by the 2005 flood. Erosive cut shows older mass-wasting deposits, covered by local derived talus sediments (reddish) and only very thin remains of the Flow 2 deposit at the canyon wall shoulder (arrows).

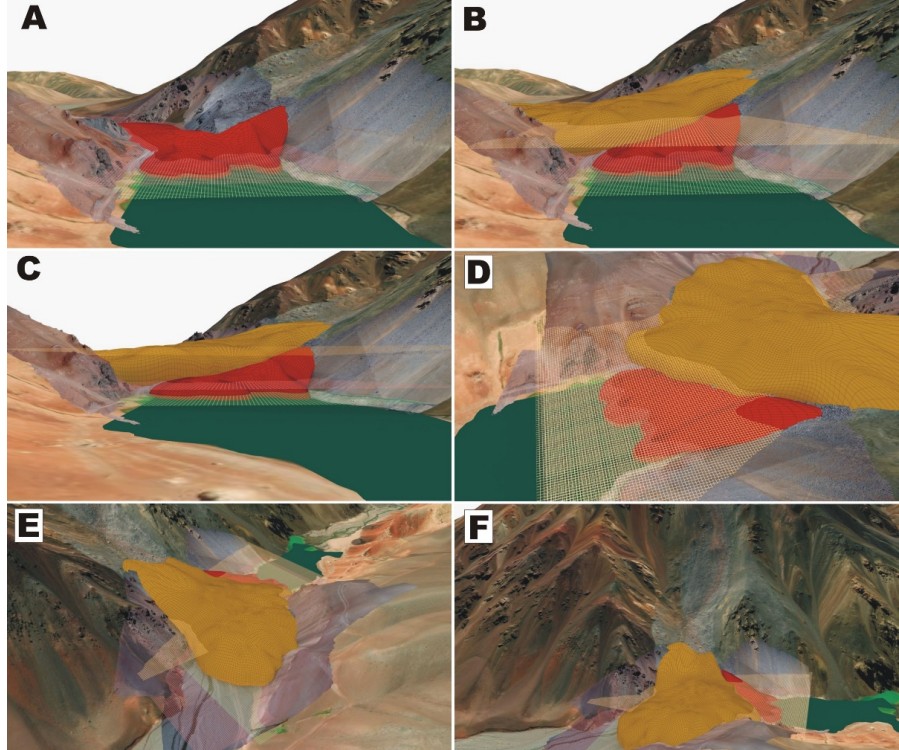

**Figure 5.** Virtual 3D recreation of the dam complex and the 3 important lake levels defined by the horizontal grids: maximum—3007; intermediate—2972; after the flood—2972; lake level now—2949 m.a.s.l. It was recreated using the Flow 1 and 2 deposits extent based on the field survey of their boundaries and the potential depositional slope for similar types of deposits. (**A**) Interpreted Flow 1 deposit (red) and observed intermediate lave level (upper grid), showing a good match. (**B**) Interpreted Flow 2 deposit (yellow) overlying Flow 1 deposit and observed upper lake level mark (uppermost grid). (**C**) Same as (**B**), viewpoint slightly below upper lake level showing that lake level would never attain the dam spill point. (**D**) Same as (**B**), viewpoint at the left side of avalanche channel. (**E**) Same as (**B**), viewpoint from downstream of the natural dam. (**F**) Same as (**B**), viewpoint from the opposite side of avalanche channel.

## 2. Description

### *2.1. Flow 1*

#### 2.1.1. Flow 1 Beginning

The deposits recognized as having been generated by Flow 1 are presently located slightly upstream with respect to those of Flow 2, so it is possible to trace them up the avalanche runway only up to almost half of the height of the avalanche channel that conducted it to the valley. Previous authors interpreted it was generated at the top of the avalanche channel that climbs from 2950 m at the river valley level to an altitude of c. 4400 m.a.s.l. [8–13]. The presence of a few eroded, cone-shaped deposits or molards [20,21] in this deposit confirm its source at altitudes above 4100 m, which is the local lower reach of permafrost bodies [14,22]. Molards are conspicuous landforms represented by isolated or grouped debris cones, each one with a surrounding skirt of a microtalus around their base, produced by the debris cone degradation. Different origins have been proposed for molards, but a recent analysis of their global occurrence, always associated with mass failures, suggested they are mostly the product of in situ melting of ice-cemented permafrost boulders [20]. Figure 6 includes sketches molard formation based on field studies and laboratory experiments carried out by one author (J.P.M.). The fact that the flow was generated in January along with the summer peak temperature led to the consideration of permafrost melting. Baldis et al. [14] linked this flow to permafrost degradation, but without providing any physical proof, such as the molards we observed in the deposit and along the runway.

#### 2.1.2. Flow 1 Transport

Few diagnostic features have been preserved except for some elements that are still present in the marginal parts of the path of this flow. Missing diagnostic features, such as levees, substantiate the Flow 1 transport process as a rock avalanche.

#### 2.1.3. Flow 1 Deposit

(a) Morphology: The top of this deposit was affected either by Flow 2 or by wave action of the growing lake, so its original characteristics are unclear. Parts of the deposits that can be associated confidently with Flow 1 show a chaotic nature, but the presence of lobes can be recognized (Figures 2 and 7). Lobe development suggests the flow became organized during its final deposition and that was not very chaotic, as lobe shifting is a well-known process to progressively fill the available accommodation space. The frontal angle of the best-preserved lobe (Figure 7) is approximately 15°, and "outrunners" were observed no more than two meters away from the lobe edge. The best-preserved lobe fronts do not show any signs of post-depositional remobilization, such as rill-marks, or micro-landslides, despite having been submerged for at least six months before being affected by the violent discharge of the Erizos Lake.

(b) Texture: The observed part of the deposit is made of very angular blocks which are clast-supported, with extremely poor sorting. Although the mean size of these very angular clasts is approximately 30 cm, there are many boulders greater than 1 m. Boulders up to three meters in maximum diameter were observed along the transport pathway. Over some of the boulders and in the inter-boulder spaces, remains of a diamictitic matrix could be identified, composed of clasts of up to 5 cm, sand, silt and clay (Figure 7C). In general, this matrix is not observed in exposures facing the lake (Figure 7D), which could be an effect of wave action winnowing the finer grains of the deposit.

(c) Source: in addition to the angular rock fragments of all sizes up to a maximum of three meters in diameter, some molards were recognized (Figure 7), which imply a minor contribution from ice-cemented permafrost [20,21]. In some cases, these ice-cemented blocks were formed almost directly with the fragmented rocky substrate, so their disintegration did not generate the typical conical molards but were observed as groups of disaggregated rocks that protrude from the deposit.

### 2.1.4. Flow 1 Post-Deposition Alteration

As mentioned above, the deposits corresponding to Flow 1 underwent the elutriation of the finer-grained fraction in the up-valley side, suggesting some wave action along the ascending lake shore. This was not observed in the top of the remaining deposit that was dissected by the evacuating flood forming a wide channel, with c. 10 m-deep scouring of the Flow 1 top. Along this evacuating channel, several inclined surfaces can be reconstructed (Figures 4 and 7), suggesting a progressive incision along with immediate deposition at the canyon exit. Flow 1 top coincides roughly with the most significant intermediate lake level mark on the valley sides (Figures 4 and 5). Parts of the Flow 1 deposit slid downslope a few meters following the lake draining events, as suggested by concave slip surfaces along the upstream sector of the dam (Figure 4).

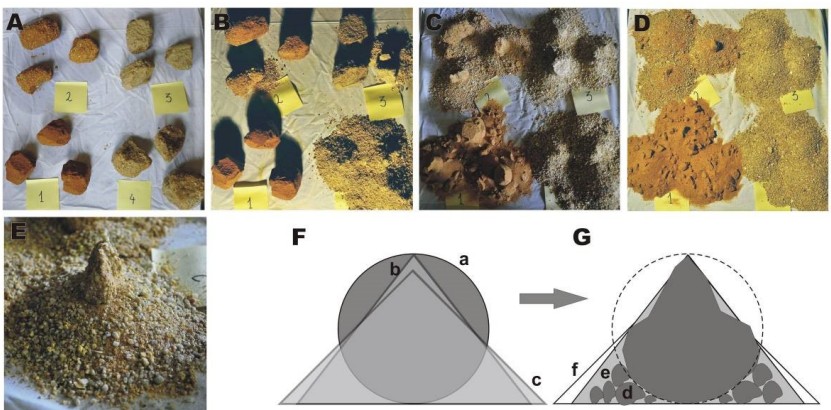

**Figure 6.** Formation of molards. (**A**–**D**) Evolution of ~8 cm large frozen ground blocks of different composition (1—fine sand; 2—fine, medium and coarse sand; 3–4—granule-rich medium–coarse sand), taken at about 6 h intervals. (**E**) Detail of a molard after 24 h. (**F**–**G**) Schematic evolution of an isometric ice-cemented conglomeratic boulder whilst defrosting; the initial sphere (a) would degrade into a cone of 51.9° (b), but will adopt the maximum repose angle of 45° (c). On the right, the intermediate molard is shown. Initially chaotic fragments are released (d), that end by forming a high-sloping talus (e), that will evolve into a less-sloping talus (f). The original boulder (dark grey) might be exposed for a short time until it degrades to form the cone apex. Modified from Milana [20].

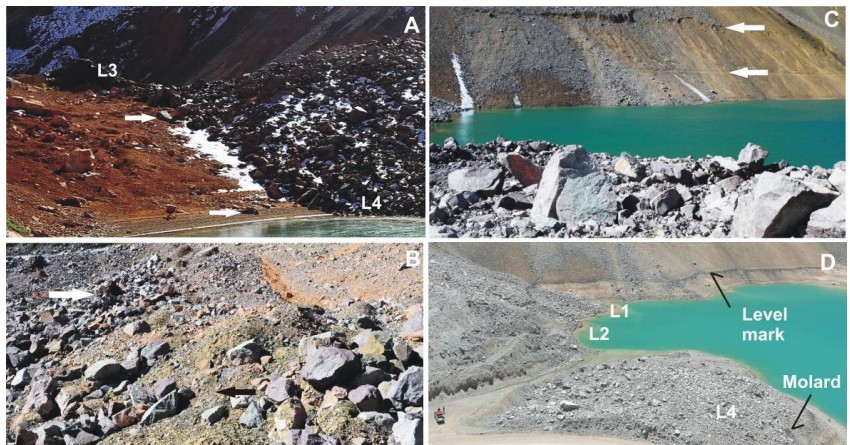

**Figure 7.** Details of Flow 1 deposit. (**A**) Distal-preserved front of Flow 1 showing two lobes (L3 and L4) with their frontal slopes and a few outrunners (white arrows). (**B**) A view of the texture of the deposit showing patches of matrix-rich debris (black arrow), while in other sectors it is almost absent (white arrow). (**C**) Outlook of the chaotic rocky deposit on top of L3. Largest boulder in the foreground is 1.5 m diameter. (**D**) Present day drained lake and altered dam showing part of the road and lobes (L1 and L2), also shown in Figure 2. Note the remaining molard and the preserved water level mark created after 16 years of stable lake level.

Remarkably, no post-depositional alterations of Flow 1 deposits were observed in its most distal parts, where deposits are thinner. Distal depositional parts are usually topographically lower, making them the logical place where a natural dam should start being eroded, at the spill-over point. However, the only important alteration surveyed over the Flow 1 deposits is the linear erosion (the canyon) attributed to the evacuation flow of the dammed water volume. This canyon now splits Flow 1 deposits into two disconnected parts, whose tops topographically match from one to the other side of today's Santa Cruz River course. The bed of the incised canyon is today floored by the coarsest boulders of the deposit that were lagged behind the outburst flood, creating something similar to the well-known hydraulic pavement. So, before this deposit was excavated artificially, it was a very stable riverbed, justifying the fact the lake size remained the same for 15 years.

*2.2. Flow 2*

2.2.1. Flow 2 Beginning

The flow marks observed along the avalanche channel in coincidence with the lower deposits, and the presence of molards all along the runway of Flow 2 (Figure 8), indicate it was sourced partially from permafrost. This flow seems to have originated in the area of extensional fractures that occurs near the upper limit of the source area, at 4400 m.a.s.l. (Figure 1). There is a slightly different colouration of this deposit that extends through the avalanche channel until its top, which allows differentiation of this flow path from that of Flow 1. The area of initiation of Flow 2 is thus 300 m above the minimum height of permanent freezing. It is likely that the freeze–thaw regime may have played an important role in opening or holding open the observed tensile fractures in the source area (Figure 1), and thus on initiating or adding mass to landslides moving down this avalanche channel. Presently, there is a sector at the source area (initial part of the avalanche channel) that has subsided c. 35 m and tensile semi-circular fractures are bounding it. We estimate that about 9,800,000 m$^3$ of rock, debris and permafrost is enclosed in this potential landslide reservoir. Penna et al. [13] made calculations of potential future failures at this upper scar finding comparable volumes. The listric cracks that may allow a new failure are easily seen in most satellite images. Thus, it is a matter of time and the degree of permafrost degradation until the next large landslide at this locality is triggered.

2.2.2. Flow 2 Transport

Based on their distinguishable colouration, deposits from Flow 2 seems to have occupied a very large area, leaving only a narrow strip original Flow 1 deposits along the up-valley side of this avalanche channel. Since Flow 2 marks are superimposed to Flow 1, the transverse extent of Flow 1 cannot be properly constrained. Based on the distribution of its deposits, Flow 2 appears to have been wider, while reaching higher altitudes over the opposite valley side (Table 1). We estimate the width of the downstream transport of Flow 2 to be close to 300 m, based on the colour marks at the intermediate altitude of the avalanche channel which match the lower deposit. Given the topographic design of the area shown in Figure 2, it seems Flow 1 deposits were contained by the pre-existing canyon formed along the Santa Cruz River, over the pre-existing mass-wasting deposits. On the other hand, deposits surveyed over the distal incised canyon shoulder are only ascribed to Flow 2 on the basis of their character.

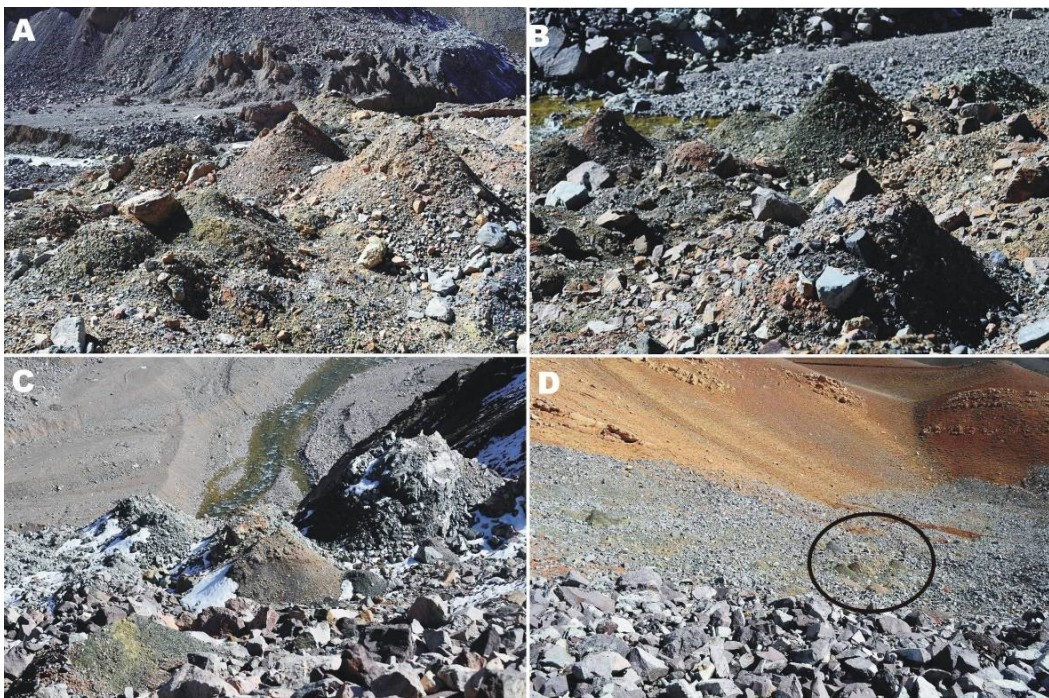

**Figure 8.** Molards (debris cones) produced by the melting of the ice-cement boulders in the dam. (**A**,**B**) Molards developed within the evacuating channel, from lagged ice-cemented boulders left by the outburst flood. (**C**) Molards located within the avalanche channel (see details below). (**D**) Molards present over the deflated and thin Flow 2 deposit, that drape the bedrock at the opposite valley margin (distal end).

### 2.2.3. Flow 2 Deposit

(a) Morphology: The final deposit does not have its own morphology but replicates the underlying topography of the previous flow, and of the pre-existing valley topography where the deposit extends beyond Flow 1 over the opposite valley margin. Thus, this deposit is a laminar sheet of high roughness, with an average thickness of less than one meter, thinning out towards the distal tip. This deposit is so thin that, in many cases, some post-depositional rilling reveals the pre-existing substrate, formed in its distal parts by Mesozoic sandstones and reddish siltstones. As a laminar deposit, it does not show a three-dimensional morphology as lobes do. However, the distal boundary of the area covered by this sheet suggests the original deposit was formed by broad lobes, larger than Flow 1 lobes (Figure 2). Based on the wider curvature defined by its forehead, the lobe that reaches the maximum height on the opposite margin of the valley seems to be the largest. More lobes can be recognized up- and down-valley, but they are smaller and at lower topographic levels (Figure 9A).

Towards the central area of the deposit, it is possible to observe places where the thickness changes rapidly by a few tens of cm (Figure 9B). These changes may indicate the original presence of internal lobes. In the down-valley areas, this deposit tends to be more homogeneous and with a more linear edge, suggesting that the residual flow was diverted to run parallel to the valley margin. It is very difficult to separate this flow from pre-existing ones due to its laminarity and almost similar composition to the Flow 1 deposit, but a detailed inspection shows that the lack of interstitial fines is diagnostic of Flow 2 deposits. In all surveyed places, remains of Flow 2 rest directly over the pre-existing topography, except along the incised canyon. This canyon existed before 2005, and was carved by the Santa Cruz River into a mixture of the Mesozoic outcrop and remains of deposits from previous gravitational flows of Quaternary age. It was later heavily reworked during the outburst flood of 2005. Sections of Flow 2 deposits can be seen over the shoulders of the canyon that provides fresh cuts showing the laminarity and internal texture of this deposit

(Figure 9D). Over this shoulder, remains of Flow 2 are present up to 3009 m.a.s.l. over the valley margin. These remains do not show any indication of post-depositional flow over its top.

(b) Texture: Unlike Flow 1, this deposit shows larger textural differences between the proximal and distal parts. In the distal parts, no blocks or molards over one meter (original diameter) have been observed.

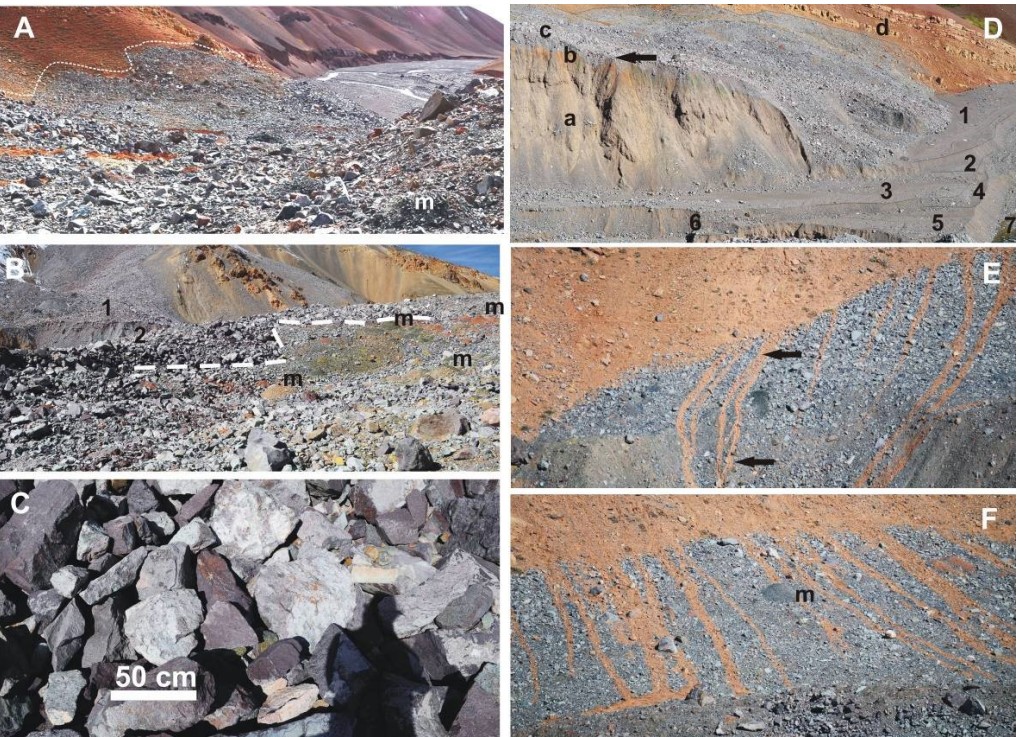

**Figure 9.** Details of Flow 2 deposit. (**A**) Lobate boundary of Flow 2 at its more distal reach. Note the sheet-like shape and sparse molards (m). (**B**) Minor changes on the thickness of Flow 2 (hatched line) suggest the existence of internal lobes. It also shows the irregular distribution of molards (m) across the deposit. Numbers 1 and 2 point to the shoulders of the pre-existing incised canyon that Flow 2 would have completely filled up. (**C**) A view of the lack of fines and lack of effect of abrasion on boulders of Flow 2 deposit. Scale bar is 50 cm. (**D**) A view of one of the canyon walls (marked 1 in (**B**), showing it was excavated over previous mass-wasting deposits (a) partially covered by reddish scree (b), derived from the sedimentary Mesozoic units (d), and topped by the thin deposit ascribed to Flow 2 (c). Note also the depositional terraces formed by the outburst flow, numbered 1–6, with 7 being the 2012 river level. (**E**,**F**) Views of the distal end of Flow 2 deposit where it is abnormally thin, as shallow rilling exposed the underlying Mesozoic red beds.

In the primary river course, the maximum size reaches c. two meters (Figure 8), while in the proximal parts (before the canyon proximal wall), the largest molards suggested permafrost boulders of up to six meters in diameter (Figure 10). Moreover, we observed a few rock fragments of up to four meters. This deposit does not present a matrix of any kind, except in the places where molards are observed, as they supplied finer-grained fragments when degraded. The outstanding rock fragments tend to be more angular and larger than those from Flow 1, suggesting a lower effect of clast interactions and abrasion during transport for Flow 2.

(c) Source: Except for two significant differences, the rock composition is essentially similar to the other deposit. Firstly, Flow 2 deposits tend to be slightly darker. The second is that this flow contains a larger proportion of molards than in Flow 1 deposit.

### 2.2.4. Flow 2 Post-Depositional Alterations

This deposit covers the pre-existing topography as a thin, laminar sheet. Considering the large boulders observed, we cannot explain the transport of this deposit with traditional transport agents, such as water, mud, air and their mixtures. Thus, the most likely carrier is snow or firn, which later melted out. The reconstruction of the possible original volume of Flow 2, based on traditional depositional angles of debris–snow avalanches, suggest this deposit lost up to c. 90% of its original volume.

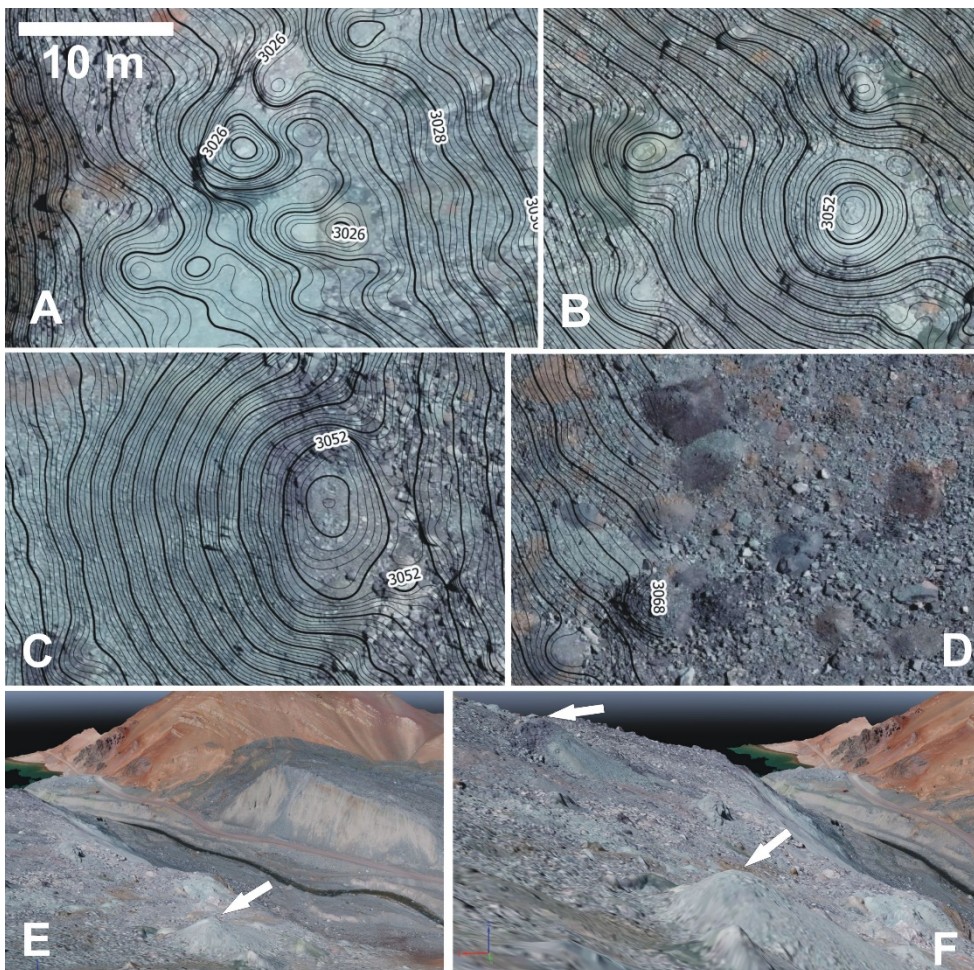

**Figure 10.** Detailed topography showing the 3D expression of molards. Location of (**A**–**D**) are shown in Figure 2. Main contours every 1 m, smaller every 20 cm. (**A**) Shows a rocky molard (centre of caption) that did not completely degrade into a cone. Note the slope change at the edge of the canyon and the easy identification of molards when they do not show chromatic contrast, as in (**D**). (**B**,**C**) Two of the largest molards that created colluvial microtalus aprons c. 10 m high downslope, and only 2 m up the general slope. (**D**) Caption at the limit of the survey showing a wide range of colours denoting a mixed composition of molards. (**E**,**F**) Virtual recreation of molards using the DEM and the surveyed photomosaic.

Along the corridor carved by the evacuating flood, deposits of this flow were completely removed. Several molards were surveyed along the flood-eroded corridor, suggesting some large permafrost boulders were left by the flood along its path (Figure 8). Since the ice of permafrost boulders of Flow 1 would have melted during the summer, we attribute those molards to Flow 2. Over the intact parts of this deposit, including most of the distal areas passing the incised canyon, no evidence of post-depositional flows was detected, and the total absence of any matrix is clear evidence of the origin of this deposit as a debris-laden snow avalanche.

## 3. Interpretation

### 3.1. Flow 1

The interpretation presented here of the processes associated with Flow 1 is very similar to those previously published. However, previous authors all considered the deposits to belong to a single flow [9,11,12,14]. It is worth clarifying that Flow 1 must have started as a rockslide (landslide), because the type of scars at high altitude, and the presence of molards indicate that the initial failure involved a mass of ice-cemented debris. Therefore, it began as a coherent movement, probably rotational, of a large mass of different materials. Large, high-speed landslides which transform into rock avalanches during transport are called sturzstrom [23,24] and show an index H/L < 0.6 [25–28], while Flow 1 has an H/L = 0.46 with a fahrböschung of 26°. Based on the publications of Davies et al. [29,30], Pollet and Schneider [26] concluded that "the mechanism of dynamic disintegration of rock material in sturzstroms that transforms a rockslide to a granular flow remains poorly understood". Given the H/L, fahrböschung and some couloir segments attaining up to 42° (average slope 23°) along a total vertical relief of 1530 m, an intense fragmentation of the original coherent mass starting the landslide is ensured. In January, thawing of the active layer causes the supra-permafrost layer to have high water contents, being the likely water supplier for lubricating the Flow 1 by wet, fine-grained material. Nonetheless, the main driver of this flow was gravity acting directly on the solid materials, matching the classification of a rock avalanche. The lobes formed at the lower reach, indicate that the rock avalanche was not transported in air but by a lubricant fluid during its last stages of emplacement. In the case of transport with air, a gradation of the largest blocks would have been observed towards the distal areas, with the largest blocks at the toe of the deposit, given their highest potential energy pushing then further, as is the case for colluvial fans [20]. This was not observed here, nor were outrunners, except for those blocks that rolled a couple meters further from the terminal lobe front.

Thus, last parts of the flow seem to have been better organized than a rock avalanche, which we explain with the presence of a fluid lubricating agent. The fine-grained matrix between some blocks (Figure 7) plus the lobes' presence suggests the flow may have turned in the last stages into a sort of debris flow. Thus, Flow 1 seems to be it started as a coherent landslide, evolving into a rock avalanche, and finally into a debris flow that blocked the Santa Cruz River, creating the first stage of the Erizos Lake.

### 3.2. Flow 2

As indicated above, the characteristics of this deposit do not match traditional flow schemes, since most sedimentary depositional models do not consider snow as an important transport agent in mountain areas (as avalanches). Snow does not leave remaining evidence after the spring/summer melting season, and thus it is underrepresented in the stratigraphic record (see below). In this dam, the large areal extent of Flow 2 without a corresponding volume indicates a mixed flow type, composed mainly by snow. Our interpretation, supported by the high-resolution topographic survey (Figure 2), suggests Flow 2 was volumetrically larger with respect to Flow 1, as it occupies a larger area as well. However, the preserved volume, given by the remaining debris, is higher for Flow 1 than for Flow 2 deposits. The missing snow volume is needed to explain the highest marks at the opposite side of the valley. Since the dam collapsed in the ablation season, we expect that the Flow 2 deposit already was heavily altered due to melting.

Flow 2 is today very laminar but climbed the opposite valley margin 30 m higher than the same deposits along the valley centre. Some flows may climb obstacles, as lahars, but we do not know of any case of flow that could climb so many meters high and leave a matrix-free, open-framework deposit composed by badly sorted angular cobbles and boulders. The only plausible explanation we have come up with is a debris-rich snow avalanche. Flow 2 also shows many lobes that are comparable to the avalanche boulder tongues first described by Rapp [31] in Lapland. Lobes or tongues often show a very regular form, a markedly concave slope profile and distal parts "may continue upwards a

little on the opposite valley side". Avalanche boulder tongues were observed by various authors, as in the French Alps [32], Wyoming [33], Rocky Mountains [34], Scotland [35], Canada [36], Colorado [37], and Spitzbergen [38,39].

Hauser [40] described a catastrophic rockslide starting from ~4350 m a.s.l., turning into a "rock avalanche, and later a hyperconcentrated debris flow with at least $15 \times 10^6$ m$^3$ due to the incorporation of snow, ice and sediments" that took place on 29 November 1987 in Chile that dammed the Río Colorado tributario of Maipo River, a disaster that took place approximately on the opposite side of the Andes of our study case. Dramis et al. [41] identified "scattered ice-cemented blocks" in a debris landslide that dammed the Adda River. Blikra and Nemec [42] tackled the issue of snow-mixed flows and snow-mixed avalanches, or "aludes" (as they are called locally as an alternative to "avalanche"), and pointed out the lack of detailed sedimentological studies of the products of these flows as follows: "Surprisingly, snowflows have drawn little sedimentological interest, although they are known to carry often abundant rock debris". These authors made a very extensive revision of snow-flow types, initiation, rheology, deposition and preservation in the geological record, so we refer to that work for a better understanding of these snow–debris mixed flows. However, we observed in the Erizos dam deposits a couple of significant differences. In our experience, most local snow-flows incorporate debris material during transport given the snowpack tends to be thin. In the study case, the flow might actually be triggered by the underlying soil and rock failure, instead of a failure in the snowpack, as it is usually triggered a snow avalanche. Evidence of this failure is the extensive presence of permafrost boulders that were evidently sourced at the apical part of this mixed avalanche. Instead, Blikra and Nemec [42] stated "snowflow transport debris that has accumulated on the snowpack due to rockfalls and related processes, including wind-blown fines; debris that has been removed from the mountain slope/ravine and incorporated en route by the flow; and debris that has been swept by the flow from the apical part of a colluvial fan". Thus, we concur with statements indicating that snow avalanches are one of the principal agents of debris transfer on steep slopes in mountain environments [43], but we believe arid realms imprint different characteristics on this process, which demands more studies. It is however important to differentiate these snow flows, from reported ice–rock debris avalanches, mostly related to glaciers [44,45], as ice will behave as a rock fragment in most cases.

In case of the Erizos Flow 2, the presence of many permafrost blocks or frozen active layer fragments, evidenced by the numerous molards (Figures 8 and 10), indicate that the collapse of a large slab of frozen ground initiated the snow avalanche. Slope instability due to permafrost degradation was investigated by various authors [14,41,46–48]. Perhaps the additional load of the snowpack on the soil fostered the collapse, as 2005 was the year with the highest snow precipitation within the last 20 years, as shown by our revision of MODIS/Terra imagery for this area. All evidence indicates that Flow 2 was initiated as a minor landslide and involved a thick snow mantle on the way. Landslides on snowpacks show remarkably large travel distances with low turbulence in the displaced mass, high velocities and momentums and collect snow on their runway, while a small percentage of the total snow content melts due to friction heat [49–51]. Snow content increases volume, water content increases entrainment of material from the path of the landslide and snow lowers the basal friction [52]. Schneider [51] investigated the "frictional behavior of granular gravel-ice mixtures" in a laboratory and observed that the bulk friction coefficient linearly decreases with ice content, and that the transformation from a dry granular flow to a debris-flow-like movement occurs when the ice content exceeded 40% by volume. This explains why Flow 2 built lobes in a fashion similar to a debris flow.

The conditions of an extensive and heavy snow cover that promoted Flow 2 also helped to mask it in available 30 m-resolution satellite images, in which it was undetectable. Indirect evidence of a taller dam is the lake level marks above the height of Flow 1 deposit, and the two-step growth of the lake area (Figures 3 and 6). Since the dam is difficult to identify in satellite images contrary to the lake, remote sensing monitoring systems would

need to focus on automatically tracking water bodies. Only the San Juan River basin involves an area of c. 20,000 km$^2$, over which these processes may occur at almost any place. An extensive literature review [42], plus our personal experience with snow-mixed avalanches, indicates that these flows often form terminal lobes, but there is no mention of snow-mixed avalanches climbing obstacles or leaving a deposit steeply sloping against the transport direction, and with blocks up to 1 m in diameter being deposited high on the opposite valley side. All evidence suggests, therefore, that Flow 2 was a snow-dominated, debris-laden avalanche.

The second difference between our observations and those of Blikra and Nemec [42] relates to the post-depositional alteration. We did not observe any finer-grained deposits developed on top of the debris deposit with stream flow evidence associated with the release of meltwater (Figure 9C), as suggested by those authors. A third difference we found was that the final thickness of Flow 2 exceeded "one boulder or cobble thick" [42]. We ascribe this difference to the landslide origin of Flow 2.

The absence of post-depositional alterations by melt-out streams over the Flow 2 deposits may be the result of the larger proportion of debris in this avalanche, with respect to those traditionally studied from more humid regions [42]. Personal observation of comparable flow deposits in the arid Andes indicates that, as melting advances, a continuous debris cover develops, making it look like a debris-covered glacier. This debris cover, overlaying the debris-rich firn, may have two effects on the post-depositional evolution of the deposit: (a) it reduces the solar energy that is the main source of ablation locally [53], slowing down ablation respect to exposed snow; and (b) it creates a permeable layer where the melted water can drain by percolation and not by surface flow, explaining the absence of stream marks on top of Flow 2. Our observations underline the need for more research on mixed snow–debris flow processes and products.

The formation of a continuous debris blanket over the Flow 2 deposit, over the areas which were not removed by the outburst flood, may explain why the non-specialist officers that visited the site several days after dam collapse did not report the existence of a debris-covered firn at the dam site. This "unknown" clarifies the need of a comprehensive study soon after the event. Detailed photographic documentation would have been extremely useful, but that opportunity was not used. This is why we insist on an early analysis of any natural disaster, undertaken by a team with specific and appropriate expertise.

Besides the abovementioned differences, the Flow 2 deposit is very similar to those described by Blikra and Nemec [42], with the addition of the permafrost boulders that turned into molards. The general appearance of the deposit is as if rock fragments had rained down, but without evidence of impact, due to the fact they were delicately deposited by melting down the original snow matrix, creating the open framework for this poorly sorted deposit [54]. Where molards formed, they released fine-grained material to occlude the inter-granular void spaces. These deposits are very frequent in the arid Andes and numerous studies [55–59] show that these mixed snow–debris avalanches are the main mechanism for the rapid formation of dynamically active [7] debris-covered glaciers. Arid-region snow flows might carry much larger amounts of debris due to the high degree of rock fragmentation fostered by the local climate.

## 4. Discussion

### 4.1. The Dam Collapse

The Erizos Lake did not exist before 2005, and researchers only mention one dam-forming landslide/rock avalanche event in January 2005. On the other hand, we supplied evidence for the existence of a second flow, of different characteristics, that would have been the one that primarily collapsed, causing a drag effect on the top of the initial dam-forming landslide. Other authors do not completely coincide with the collapse of the dam, most mention that rupture was by lake water overtopping with headward erosion [8,10–13], overtopping by tsunami [9,10], slope instability [10], seismic shaking [10] and a direct hit by a rock mass [14]. However, none included detailed field studies of the collapsed dam.

Besides, several witnesses of the flood front near the dam location, and the report of its advance along the drainage network on local news, support a sudden collapse. Thus, the question is whether the sudden collapse of this dam was possible.

On-site additional evidence of the collapse is given by the sedimentary accumulation terraces formed immediately downstream of the dam. Figures 8C and 9D show that the collapse generated several depositional terraces (six are depicted in Figure 9D), with the upper one having an altitude of 2950 m, while the lake level before being emptied was 3007 m, ending at 2957 m, and the dam closure point was 3009 m for Flow 2 (Figure 2). For the preserved part of Flow 1, the closure point was 2968 m, but it might have been higher within the incised canyon, to enable the lake level to attain the intermediate mark of 2972 m.

Since the closure point was 3009 m.a.s.l., and the maximum lake level was 3007 m.a.s.l., the dam could not have been overtopped unless the dam height decreased along the valley centre line. Dam compaction is unlikely due to the scarce matrix and boulder-like nature of the deposit; although, permafrost block melting or snow melting would be possible. However, overtopping would have created initial sediment-laden flows, generating a high-sloping fan downslope of the dam. Subsequent backward erosion, lowering the dam height, would create successive incised accumulation terraces, with progressively shallower slopes with lower altitude, when projected towards the dam crest. This was not the case, as the first flood terraces were only 2–3 m higher than the final river course and there is no evidence of the hypothetical overtopping. The first depositional terrace occured at c. 2950 m.a.s.l. and expanded all the valley width after the exit of the incised canyon, covering passively the more distal deposits of Flow 2.

Any collapse by overtopping would need to explain the lack of any depositional elements of those flows at higher elevation than 2950 m.a.s.l., when the potential breach would have started developing 57 m higher. Initial flows would have been more capable of transporting sediment given the initial slope was higher, but there is no evidence of them, while there is overwhelming evidence of the lower topographic flows. Another important aspect of evidence to reconstruct this collapse is given by the witnesses that were observing the flood, a few km downstream of the dam collapse. They indicated that the river was flowing, suggesting "normal" conditions upstream, so it is difficult to think that headward erosion would suddenly increase to cut down the entire dam in a short period. These depositional terraces support, therefore, the sudden collapse of the dam along the incised canyon only.

Other elements that play a role are (a) the fact the dam collapse affected the upper part and not the whole dam; (b) the fact this was the only one outburst flood known in this region; (c) the meaning of the significant, intermediate, water-level mark. These three elements fit with the interpretation we supply here: the dam was made up by two flows of different characteristics, and only the second deposit suddenly collapsed, whereas the earlier one was only eroded at its top. Our interpretation that the deposit of Flow 2 was mainly made of snow explains the sudden collapse, similar to a jökulhlaup, where an icy dam is simply floated by buoyancy and moved downstream along with most of the lake volume. This explains why there are no traces of evacuating flows above final lake level of 2957 m.

Even if it was a sudden rupture, the exact process of collapse is a matter of debate, since there is no information about the last stages of the dam/lake complex evolution. Due to the stabilised water level at 3007 m.a.s.l. (Figure 2), a long-lasting, steady-state lake charge and discharge is suggested. Both Flow 1 and Flow 2 were superimposed over pre-existing debris dams that were incised before 2005 and formed a canyon elaborated over these colluvial deposits (Figures 2 and 9). Therefore, it is highly likely that the material underlying the 2005 dam was quite porous and allowed a significant amount of water to pass through it. This possibility is consistent with the fact that the lake's maximum level never reached the spill point at 3009 m, but seems to have been stable at 2–3 m below the spill point of Flow 2, according to the 3D-UAV model (Figure 2). Given this

potential steady-state situation, the dam should not be broken unless a dynamic process was happening at the same time in the dam. We interpret that the summer ablation created the instability reducing the weight of the dam, until the weight of the dam equilibrated with the hydrostatic pressure of the water, floating a part of it. Only the central strip of the dam, coinciding with the trace of the pre-existent canyon, collapsed. Thus, it is possible that it was the dam ablation process that caused the imbalance, as illustrated by Figure 11.

Given the uneven lower topography, we can interpret that the thicker segments of Flow 2 would be more unstable as it had deeper portions submerged. A simplified scheme of the situation of an icy dam filling a deep pre-existing canyon and spreading over elevated shoulders is shown by Figure 11. Pressure on an object submerged in a fluid is $p = \rho \times g \times h$, where $\rho$ is the density of the fluid, g is gravity acceleration and h is the height of the fluid above the object. We used the arbitrary density of 0.9 g/cm$^3$ for the Flow 2 deposit, based on the estimated 10% of debris, whose density is c. 2 g/cm$^3$ mixed with compressed firn of 0.8 g/cm$^3$. This simplified geometrical array of the reality portrayed in Figure 2 shows that the pressure at the base of the deposit sealing the canyon is higher, making it the logical avenue for floating that section of the dam. Considering a 30 m-deep canyon, and that the Flow 2 deposit is 10 m thick over the shoulder, with a water level 2 m below the top of Flow 2, we can see that the water pressure at the base of deposits located over the shoulder is negative. The pressure exerted by the deposit is 88.3 kPa, and the water is 78.5 kPa at the base of Flow 2, making that scenario stable. However, the same calculation at the base of Flow 2 within the canyon, with a hypothetical 40 m-thick deposit and 38 m of water column, gives 372.8 kPa for water and 353 kPa for the deposit. In this situation, only the dam section along the canyon would be floated some time before, when the water reached the 36 m point, that equalized the pressures. The water level would need to reach 39 m to remove the dam lying over the shoulder, explaining why the laterals were untouched by the evacuating flood.

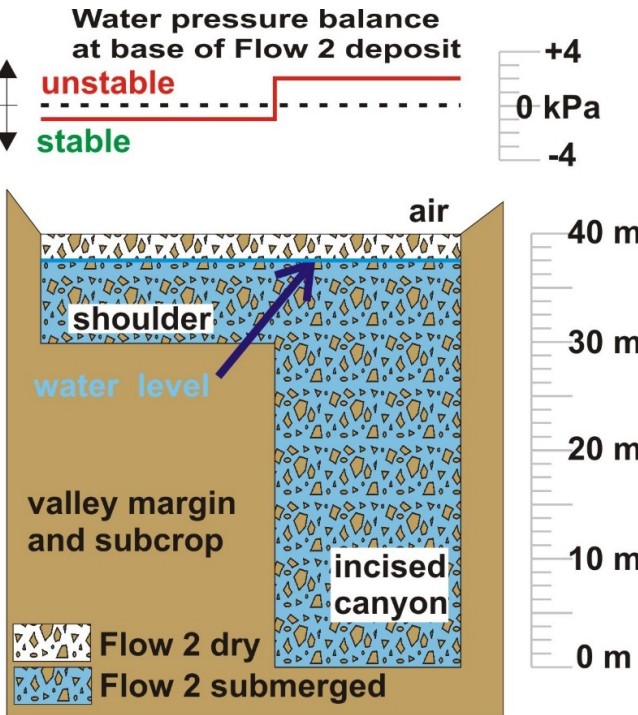

**Figure 11.** A simplified scenario to portray the uneven distribution of hydrostatic pressure along the base of Flow 2 deposits, due to the pre-existence of the incised canyon. The pressure balance between the dam and water is shown by the upper curve (arbitrary density for the snow avalanche dam of 0.9 g/cm$^3$ was used), showing that the dam over the shoulder is stable, while along the incised canyon the situation is inverse, that dam section would have been floated before reaching the portrayed scenario.

Given the likely failure happened along a narrow canyon, the erosive potential of the outburst flood was intense, causing the erosion of the uppermost part of Flow 1, cutting its top by 15 m, which is defined between the intermediate lake level at 2972 m and the level after the flood at 2957 m.

Therefore, we infer that the hazardous process was associated only with Flow 2, due to the low density of the dam created. Such dams are very common in this region, but, in general, the water works a way out at the base of the dam, so in the spring a firn bridge can be seen. This was not the case, probably due to the large volume involved that created a high pressure at the base of the deposit column, helping to make it impermeable. Either the increasing lake level or the decrease in the total weight of the dam created the instability to allow the central segment of the dam to be floated. We are inclined to see the alternative, of ablation subtracting the dam weight, as playing the main role in reaching the breaking point.

Ice vs. snow dams: It is interesting to note that, in the study region, a famous flood that also caused a large amount of damage along the Mendoza River in 1934 was also linked to a jökulhlaup [60]. In that case, it was a glacial closure, created by the advance of Del Plomo surging glacier, creating a lake along the river of the same name. That dam was made of glacial ice with a density of 0.9 g/cm$^3$ (plus potential debris), so the water had to reach at least 90% of the thickness of the seal to float it. In the case of snow avalanches, the dam density might be lower, and varies between 0.7 and 0.8 g/cm$^3$, according to the degree of compaction of the deposit. The packing of snow in an avalanche is highly heterogeneous, and more if large amounts of debris that add density are involved. Another characteristic of snow dams is that they can form at lower altitude than glacier ones in this arid setting, and hence they become subjected to higher levels of ablation, that extract significant weight from the dam on a daily basis.

For such lower densities, dams formed by snow avalanches may collapse with less filling than glacial ones, but the physical process is exactly the same: either the glacial ice or the snow are suddenly floated by the water that is denser, and not only is the water released once the dam is detached from the ground, but the entire dam itself becomes part of the flood, in a process that acquired the Icelandic name jökulhlaup, given the frequency of these phenomena in Iceland. As indicated above, the dynamic evolution of a snow-formed dam could be quite different due to the fact it tends to be more permeable than glacier ice, and that its interface with the original ground tends to be less bulldozed than the case at the base of a moving glacier, so they are prone to show more leakage. Several authors [43,61,62] reported cases of snow avalanches that temporarily impounded rivers and created lakes, with examples of catastrophic floods or mudflows caused by the collapse of snow avalanche dams [63–66]. Besides, the seasonal availability of snow can foster very large avalanches that can create significant disruptions of normal fluvial activity [43]. A recent example was the large snow avalanche that damaged the Alaskan Richardson Highway, creating a dam and associated lake that led to the creation of the term "damalanche" [67]. Most of these snow avalanche dams were produced by dense wet snow since their weight and density possess enough strength to temporarily dam creeks and rivers and create short-term lakes. It is documented that the longevity of snow avalanche dams ranges from a few hours [68] to a few weeks [69,70].

*4.2. Impacts of Alternative Interpretations on Remediation and Prevention*

This section analyses the impact of different interpretations on the management of similar future events. Thus, we compare different views of this study case to highlight the aspects which, if not properly considered, may lead to incorrect decisions. Some concerning remedial actions have been already taken.

Our field surveys helped to differentiate a second flow event, whose deposits merged with the first. This second flow was not recognized by the previous teams analysing this event, probably because no detailed field work was carried out previous to this study. We could not find any detailed photographs of the deposits composing this natural dam.

Therefore, the following elements were not considered when interpreting the dam failure: (a) a detailed sedimentological analysis of the dam deposits; (b) a stratigraphic assessment of the event sequence; (c) a geometrical correlation between different elements of the dam and the surroundings; (d) a detailed altimetry and topography; (e) a proper geocryological field assessment; (f) an analysis of the flood depositional terraces and their development along the canyon. Previous authors did not mention the molards either. Our two field surveys provide the first sedimentological characterisation of these deposits, and our UAV model (Figure 2) provides evidence to support our interpretation that Flow 2 was not a rockfall but a snow-dominated mixed avalanche. This new interpretation also allows the enigma of the sudden dam collapse to be solved, as it may have collapsed by simple hydrostatic laws, causing the catastrophic outburst flood. The outburst flow was also unstable, as several depositional terraces (Figure 9D) were formed as the evacuating stream cut down few meters into the rocky debris of Flow 1 deposit.

In order to evaluate the actions that can be taken after a disaster and to avoid or alleviate the potential negative consequences of future events such as this, it is mandatory to properly understand the circumstances of the event. We demonstrated here that a reasonable doubt about the overtopping and backward erosion hypothesis exists, since most records suggest a collapse similar to a jökulhlaup. In the following sections, we discuss the two main types of actions to be taken: (1) remediation and/or mitigation, which consider actions that may decrease the danger of future similar events occurring at the same location; (2) prevention, which are actions taken to prevent this or similar events that may occur all over this or other drainage basins.

### 4.2.1. Remediation

Soon after this mega flood occurred, the local media indicated some professionals suggested engineering to drain the remaining Erizos Lake, because they considered it posed a danger to society. Our 2012 survey (most field pictures included here are from that year), suggested the lake was quite stable, and the clear mark at 2957 m (Figure 2), made over 16 years, attest this lake is stable. Over 2021, large parts of the dam were removed and an access road crossing the natural dam complex was established, producing an 8 m Erizos Lake level drop (Figure 2). After crossing the dam, the road follows the coast of the remaining lake. We believe actions taken at the dam location may have dangerous consequences due to the fact a natural disaster happened recently, and perhaps the remediation action (deepening the lake drain to lower the lake level) may actually worsen the situation by increasing the potential danger of future events.

The artificial downcutting of the natural dam that had been stable for sixteen years may create a more dangerous scenario if another flow, such as Flow 2, rebuilds the dam. The new potential icy dam to be floated will have a root 8 m deeper along the artificially excavated channel. Thus, in a new potential collapse, the heavy material of Flow 1 will not act, restraining the erosive potential of the outburst flood. Thus, the remediation action may turn into the opposite in the hypothetical case that a flow similar to Flow 2 occurs again. We interpret it would have been better to let nature work out this problem. With time, the lake would become filled with sediment from the upper Santa Cruz River, reducing the potential volume of water that can be enclosed at this reach. At the same time, the stream draining the lake would continue to downcut through the dam, re-establishing the equilibrium slope of the river as it did before. The present incised canyon, which existed before this recent natural dam, is proof that this would be the natural evolution. It was also not dangerous to leave the lake as the flood left it, since the outburst flood created a very stable riverbed over the dam, by leaving largest boulders lagged along this path that was not modified over the last 16 years. With time, the remains of the dam would have been eroded by slow backward erosion. Once the lake becomes filled with sediment, a connection between the river up- and downstream of the dam would be produced, causing incision along the lake sediments and joining it with the incision through the dam sediments. Natural sedimentation would

have filled the dangerous volume with a solid, decreasing the potential water volume for a flood.

Another risk introduced by that access road is that it is a lake-side road. On the dam area, there is a direct impact by a rock avalanche risk, and along the lake-side road, there is the risk of the impact of a solitary wave. As mentioned above, a series of open, partially ice cemented listric fractures enclose a large rock volume of c. 11 million of m$^3$ that already descended 35 m and is hanging over the trace of the road. This location is ready to release about the same estimated volume of Flow 1 [14]. As the year 2021 was dry, the seasonal thermal wave may advance deeper into the ground, since energy will not be absorbed for melting ice in the active layer. This incomplete refreezing of the active layer over stable permafrost has been previously demonstrated by geophysical measurements in this region [71]. Thus, summer 2021–2022, or any other dry summer, may pose an extreme danger of a new landslide and a subsidiary rock avalanche due to the adverse meteorological conditions, making this road quite risky to the eventual passenger.

Besides the risk of the impact of millions of tons of rock over a potential driver, it needs to be considered that this mass may impact the remaining lake with a lot of energy after 1.5 km of vertical travel distance. The impact of such a large volume at high speed onto the lake surface may create a single solitary wave that will run directly over the opposite coast, where the coastal road was traced, destroying any vehicle. There are many cases recorded of large solitary waves created by a falling-rock avalanches and affecting the opposite side of mountain lakes and fjords (e.g., Lituya Bay, AK, USA: [72]). The Lovatnet 1934 event is one of the best studied events; see [73], and references therein. In that case, only c. 1 million m$^3$ fell 800 m into the Lovatnet Lake, generating a c. 74 m-high wave which hit the towns of Bodal and Nesdal, killing 74 people. These events are so well known that a recent movie called "The Wave" was created by the Norwegian director Roar Uthaug around the concept of a potential wave that could be formed at Geiranger fjord due to the presence of cracks that open slowly and are currently monitored. These waves rank among the highest known and have been responsible for many casualties. In Los Erizos area, a potential larger volume and a longer run-out may create a larger event than that of the Lake Lovatnet rockfall. Such a disaster could be easily prevented by using an older existing road, avoiding this dangerous spot.

### 4.2.2. Prevention

There is some overlap of remediation and prevention. However, we decided to apply the term prevention for the governance actions that could help to alleviate, prevent, or avoid the effect of a natural hazard. In 1944, almost 90% of San Juan city (Argentina) was destroyed, so an institute dedicated to the study of seismic rules for civil constructions was created. Today, San Juan city ranks among the safest in relation to earthquake intensity, giving a good example of intelligent governance. It took a disaster involving massive loss of life and property to motivate this.

In the Erizos Lake case, many expressed in the public media that a monitoring system would be deployed, but sixteen years later, no monitoring system exists. A prevention system based on the automatic revision of daily and weekly satellite images of different resolution would be easy to design and deploy at relatively low cost. This would be an essential first step to prevent future potential damages. A similar disaster to the Erizos Lake dam collapse may occur at any place in the vast San Juan River basin. This danger is increased by the impact of climate change in slope instabilities created by permafrost degradation, evidenced by the high frequency of molards in this region [20]. Besides a monitoring system focused on the new water bodies created, a second action to be taken in order to prevent future occurrences is an open call for highly trained professionals on natural disasters, in order to form a sort of council to advise the government on these kinds of events.

A third important action to be taken would be to elaborate risk maps to identify points of known dangers and others with potential danger and to make it publicly available to

prevent future accidents. With such information, the choice of dangerous road traces, such as the one through the Erizos natural dam, could be prevented.

### 4.2.3. Memory

Although being a word of an Icelandic origin, jökulhlaups may be quite frequent in the arid Andes. In 1934, a large flood affected the inhabitants of Mendoza city, Argentina. Mendoza is located just 165 km south of San Juan city, that was the economic epicentre of the Santa Cruz River flood damages. Both cities are located more than 150 km away from the closest glacier, but they were heavily impacted by these jökulhlaups. Investigations made after the 1934 Mendoza flood found a surging glacier advanced suddenly and blocked the Del Plomo River for a single season [74]. A perfect scenario for a jökulhlaup was created. A few decades after, a second closure formed a lake, but it was drained slowly through a subglacial tunnel that was kept clean until the ice mass melted out completely [75]. The 1934 event fostered the creation of the national institute for snow and glaciers in Mendoza (IANIGLA) in order to deal with dangers and hydrological issues related to the cryosphere, but they did not develop tools for nation-wide cryospheric-related danger prevention. As already said, a simple monitoring system could be deployed, but a political decision is needed for that action that may prevent some of the large economic costs that natural disasters cause. This decision is urgent.

The Santa Cruz event is perhaps not a perfect comparison with a jökulhlaup since the dam was not formed by a glacial process (jökull means glacier). However, the importance of the periglacial environment in this event is clear, given the significant presence of molards in both Flows 1 and 2 (Figures 2, 8 and 10). In the case of icy dams, it is only a matter of time to reach the level at which the dam is floated, creating the most violent scenario. The existence of two well-documented large floods ascribable to a jökulhlaup raises the likelihood of another potentially more dangerous event. The question is not whether it will happen, but when and where, and thus it is of the utmost importance to deploy strict prevention measures.

## 5. Concluding Remarks

We demonstrated here the need for a rapid collection of evidence post hoc to properly understand a natural disaster. Time changes the original evidence rapidly, precluding unequivocal conclusions that would be for the best interests of the society. While forensic geological analysis could be always helpful, it is always better to collect evidence as soon as the event occurs, as customary in criminal investigations. Our contribution also demonstrates the need for a well-educated society, with access to hazard analysis to judge whether the information produced by designated authorities dealing with the destructive event makes sense. In the case of the Santa Cruz flood, a reasonable doubt existed concerning whether the collapse was sudden or caused by progressive overtopping and rapid backward erosion. Additionally, the natural forums for discussing these issues are the peer-reviewed scientific journals that, therefore, play a fundamental role in societal improvement.

The well-known concept about immediate actions after any geohazard is mandatory due to evidence degrading with time. We may suggest the following actions: (1) An interdisciplinary team composed of specialists of the natural processes involved should be convoked, to analyse the data that was acquired by surveys, as soon and as objectively as possible. (2) If specialists are not immediately available, the complete set of evidence should be recorded by technicians, while specialist should review the field location as soon as possible, since that field survey will often provide different and usually more conclusive evidence, as demonstrated here. (3) The same interdisciplinary team should be involved in defining mitigation and prevention actions that should be forcefully taken by authorities. It is important to underline that aerial or remote sensing inspections alone are not enough to define the nature of a geohazard. The lack of field control of previous studies inhibited detection of (a) the deflated deposit of Flow 2, (b) the many molards

indicating permafrost failure, and (c) the absence of matrix in the Flow 2, that characterizes nivo-detrital avalanches.

Remediation/prevention is the lesson we should always learn after a disaster. Remediation measures taken without complete knowledge of the destructive event may lead to more dangerous situations. An example of a bad decision is the road built crossing the collapsed dam and then running along the lake coast, introducing unnecessary risks. This exemplifies the need of better governance in natural disaster matters. We hope the case portrayed here may help to improve governance actions over regions that are poorly managed with respect to natural disasters.

The lesson here is, therefore, that natural risk effects might be amplified by poor governance. It is usually a political decision to seriously consider these dangerous processes, using the highest technical standards, or to just let the time pass. The latter solves nothing, and given the advance of societies over natural spaces, coming disasters are likely to be worse. As a final thought, we would like to draw attention to the concept which Jared Diamond summarized in the tittle of his book—*Collapse: How societies decide to fail or succeed*. In the case of natural hazards, it is only a matter of governance to develop better approaches to devastating natural events that claim many lives around the world every year.

**Author Contributions:** J.P.M. designed and wrote the manuscript, was in charge of the field work, figure edition and text revision; P.G. processed the digital and satellite information, DEM and image processing, text edition and references organization. All authors have read and agreed to the published version of the manuscript.

**Funding:** This research received funding from CONICET grant PIP 01000852CO.

**Institutional Review Board Statement:** Not applicable.

**Informed Consent Statement:** Not applicable.

**Data Availability Statement:** Not applicable.

**Acknowledgments:** We are deeply indebted critical revision of the early manuscript by Ben Kneller, to government officials that granted access to the site in 2021; Ing Juan Carlos Caparros (Secretario del Agua) facilitated use of San Juan helicopter while San Juan Mining ministry Ing. Carlos Astudillo, and directors Ing. Hugo Chirino and Lic. Roberto Luna made it possible to survey the high-resolution topography of the dam remains. The Mining company Xstrata granted access and provided logistical help to carry out the first sedimentological revision of the dam in 2012. We also acknowledge the help of Susana Heredia for obtaining the drone to survey the dam. The company GHB Group provided data processing to elaborate the DEM used for this contribution.

**Conflicts of Interest:** The authors declare no conflict of interest.

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
