# Peer review of "Forensic Geology Applied to Decipher the Landslide Dam Collapse and Outburst Flood of the Santa Cruz River (12 November 2005), San Juan, Argentina"

_2624-795X, doi:10.3390/geohazards3020014_

Round 1
Reviewer 1 Report
Few comments about the manuscript are listed below.
- English level can be improved. Small mistakes here and there, but still understandable.
- Minimum reference in Description, Interpretations, and Discussion cause the manuscript to be very subjective and reduce its scientific soundness.
Author Response
Dear Editor,
We appreciate and thank you for the opportunity to resubmit this revised manuscript.
In general we accomplished reviewers editors and we did a very deep review after these new comments. All our edits can be seen in the file with changes tracked we are attaching. Some specific areas are commented below.
- We tried to follow the instructions of the reviewers but we would like to mention that given the lack of exact locators (e.g., lines were problems were observed ) we may have failed in correcting issues as the one mentioned by one reviewer “sometimes it was not clear to me whether some statements were reasonable (data-backed) reasoning or speculations. It would be best if, when you write things like "this may be caused by", you could make it more apparent whether this is a probable thing ("may likely be caused by", "it is reasonable to think that", "it is highly probable that"). Thus, we tried to reword all those sections where we observed that defect, and we hope we did a proper job.
- We also added some 3D images, whose production delayed this resubmission a little. It is always problematic to explain the 3D media experts how a true sedimentary surface may look, but we believe they interpreted our thoughts, as the reconstructed original deposit surface is a combination of sedimentary logics and lots of imagination, given the fact all surfaces were modified by ablation, breaching and slumping.
- After our edits following reviewer’s advice, we also asked a native English speaker to screen the text for awkward constructions and mistakes. We hope we fixed most of the problems, although a “flavor” of foreign English probably still remains.
- We tried to reduce the text, simplifying the Abstract from 416 to 381 words (8,4%), and the body text from 11,750 to 10,500 (10,7%). We followed reviewer’s comments to cut those parts of text indicated while in fact we may have enlarged some sections as suggested by reviewers (as focusing and simplifying the forensic concept)
- We do not think a table comparing others authors and us would be of great value given the only difference we have with all them is the existence of Flow 2, which is well detailed in the text. However, we can add that suggested table. In such a case, it would be welcome to know which variables we should add on that table. As indicated, the only difference is the existence of flow 2
- References One reviewer observed; “Minimum reference in Description, Interpretations, and Discussion cause the manuscript to be very subjective and reduce its scientific soundness”. We tried to arrange this problem as far as we can without making a very extensive reference list (there are 74 refs). Important to say is that many articles referenced are reviews (so we use “and references therein”), in order to avoid extensive references that may not specifically apply to this case. As the reviewer did not indicate specifically where the problematic points are in the MS, we did our best to solve this defect. We hope we improved this area largely, as we tried for point 1.
- Figures. We followed the indications of reviewers as follows. Fig. 3: we added on the lake area plot, the % of snow over the area of the dam (c. 20 has), as it shows how the lake area that was stable increases in coincidence with the maximum snow cover of the dam. We also added error bars for the surface measurements. and we changed layout. Fig.: 10 (now 11): we improved it, but we need to remark it is just a sketch to show how the basal hydraulic pressure changes according to the depth of an icy deposit submerged. Fig. 5: this one is new, moving following numbers one up. We thought this would be the best place for the 3D recreation that a reviewer suggested to include. Fig, 10 (older Fig. 9), was also modified to include virtual views of the molards (a recreation using the DEM with the photomosaic overlaid and placed in 3D perspective. Unfortunately, as we used the true surface (taken form orthogonal view) when it is draped over an inclined surface, it is stretched so some areas do not look realistic. This can be solved by adding artificial textures, but we chose to leave the true surface image captured by the drone.
- Given we already produced the 3D recreation, using our detailed 3D survey overlaid the Aster GDEM (lower resolution), I was informed that an animation of the breach process could be supplied, for the video abstract. Please advice when this content would be supplied as it may take a few days to adjust that animation.
Reviewer 2 Report
In this interestingcontribution, the authors use a well-known case of landslide dam collapse in Argentina to argue on the need and benefits of a "forensic geology" aproach for the study of disasters. I think this perspective is interesting and useful for those doing research on such disasters and/or working on their prevention and mitigation.
I do not argue nor take position in the debate presented by the authors, but I do find their arguments "well argued". I do have, however, some contents concerning the presentation of the work, which I think could be improved to become more effective in the eyes of the readers.
The manuscript contains some beautiful pictures but also some, let me say, ugly graphics. This inconsistency strikes the eye and I suggest that the authors really improve the visualisation of figures 3 and 10.
Moreover, I feel that some sketches (2D or better 3D visualisations) are necessary both in the description and in the interpretation sections to make the processes and mechanisms the authors describe more visually understandable. I think this would really increase the impact of this paper.
Also, concerning the existing and contrasting views on the event and the subsequent countermeasures, the authors could include a table to summarise the positions and where the authors stand.
About the language used, sometimes it was not clear to me whether some statements were reasonable (data-backed) reasoning or speculations. It would be best if, when you write things like "this may be caused by", you could make it more apparent whether this is a probable thing ("may likely be caused by", "it is reasonable to think that", "it is highly probable that") or a minor probability or even an idea that is not demonstrated. I hope you got what I am trying to say.
Finally, I think the abstract is a bit long (and discouraging) and perhaps you could leave out some numbers and summarise them in text form. The introduction also could be a bit shortened and better structured. Personally, I would focus on the "forensics" concept in the introduction and on why you chose the case study, then have a separate section where you describe it (but not describe it in both sections). This is just my two cents. I really found the manuscript interesting overall.
Author Response
Dear Editor,
We appreciate and thank you for the opportunity to resubmit this revised manuscript.
In general we accomplished reviewer’s indications and we did a very deep review after these new comments. All our edits can be seen in the file with changes tracked we are attaching. Some specific areas are commented below
- We tried to follow the instructions of the reviewers but we would like to mention that given the lack of exact locators (e.g., lines were problems were observed ) we may have failed in correcting issues as the one mentioned by one reviewer “sometimes it was not clear to me whether some statements were reasonable (data-backed) reasoning or speculations. It would be best if, when you write things like "this may be caused by", you could make it more apparent whether this is a probable thing ("may likely be caused by", "it is reasonable to think that", "it is highly probable that"). Thus, we tried to reword all those sections where we observed that defect, and we hope we did a proper job.
- We also added some 3D images, whose production delayed this resubmission a little. It is always problematic to explain the 3D media experts how a true sedimentary surface may look, but we believe they interpreted our thoughts, as the reconstructed original deposit surface is a combination of sedimentary logics and lots of imagination, given the fact all surfaces were modified by ablation, breaching and slumping.
- After our edits following reviewer’s advice, we also asked a native English speaker to screen the text for awkward constructions and mistakes. We hope we fixed most of the problems, although a “flavor” of foreign English probably still remains.
- We tried to reduce the text, simplifying the Abstract from 416 to 381 words (8,4%), and the body text from 11,750 to 10,500 (10,7%). We followed reviewer’s comments to cut those parts of text indicated while in fact we may have enlarged some sections as suggested by reviewers (as focusing and simplifying the forensic concept)
- We do not think a table comparing others authors and us would be of great value given the only difference we have with all them is the existence of Flow 2, which is well detailed in the text. However, we can add that suggested table. In such a case, it would be welcome to know which variables we should add on that table. As indicated, the only difference is the existence of flow 2
- References One reviewer observed; “Minimum reference in Description, Interpretations, and Discussion cause the manuscript to be very subjective and reduce its scientific soundness”. We tried to arrange this problem as far as we can without making a very extensive reference list (there are 74 refs). Important to say is that many articles referenced are reviews (so we use “and references therein”), in order to avoid extensive references that may not specifically apply to this case. As the reviewer did not indicate specifically where the problematic points are in the MS, we did our best to solve this defect. We hope we improved this area largely, as we tried for point 1.
- Figures. We followed the indications of reviewers as follows. Fig. 3: we added on the lake area plot, the % of snow over the area of the dam (c. 20 has), as it shows how the lake area that was stable increases in coincidence with the maximum snow cover of the dam. We also added error bars for the surface measurements. and we changed layout. Fig.: 10 (now 11): we improved it, but we need to remark it is just a sketch to show how the basal hydraulic pressure changes according to the depth of an icy deposit submerged. Fig. 5: this one is new, moving following numbers one up. We thought this would be the best place for the 3D recreation that a reviewer suggested to include. Fig, 10 (older Fig. 9), was also modified to include virtual views of the molards (a recreation using the DEM with the photomosaic overlaid and placed in 3D perspective. Unfortunately, as we used the true surface (taken form orthogonal view) when it is draped over an inclined surface, it is stretched so some areas do not look realistic. This can be solved by adding artificial textures, but we chose to leave the true surface image captured by the drone.
8. Given we already produced the 3D recreation, using our detailed 3D survey overlaid the Aster GDEM (lower resolution), I was informed that an animation of the breach process could be supplied, for the video abstract. Please advice when this content would be supplied as it may take a few days to adjust that animation.
Round 2
Reviewer 1 Report
This manuscript has been improved from previous versions
Author Response
Dear Editor,
We appreciate and thank you for the opportunity to resubmit this revised manuscript.
We tried to follow all the reviewer’s last indications. With time passed and after the re-reading if the MS, we strongly agree that the last parts were a bit emotional. Therefore we tried to remove any subjective section out of the text.
We also like to draw the attention on the fact the annotated pdf uploaded the reviewer(s) (geohazards-1601692-decision.v3.pdf ) was not including the updated figures. We do not have an idea of how did it happened, but the fact is that Figure 3 and 10 (now is 11) were the older versions, and in fact the new added Figure 5 (digital recreation of the closure) was also not present in that pdf. Perhaps there was a mistake in our uploading, so I am changing the name of the old files with the track of changes in order to prevent that problem.
In any case, it become clear a new revision was required as besides the mistakes identified by the reviewer(s) we identified an equal or larger number of mistakes. Probably is not a perfect product now, but we believe is much more improved now.
I hope you considered the manuscript is acceptable in the present shape.
Sincerely yours,
Juan Pablo and Philipp
Reviewer 2 Report
I appreciate that the authors took my observation in serious consideration. I think the manuscript has improved significantly and I see no issues preventing its publication. Once again, I recognise that this is an interesting and valuable work.
Author Response
Dear Editor,
We appreciate and thank you for the opportunity to resubmit this revised manuscript.
We tried to follow all the reviewer’s last indications. With time passed and after the re-reading if the MS, we strongly agree that the last parts were a bit emotional. Therefore we tried to remove any subjective section out of the text.
We also like to draw the attention on the fact the annotated pdf uploaded the reviewer(s) (geohazards-1601692-decision.v3.pdf ) was not including the updated figures. We do not have an idea of how did it happened, but the fact is that Figure 3 and 10 (now is 11) were the older versions, and in fact the new added Figure 5 (digital recreation of the closure) was also not present in that pdf. Perhaps there was a mistake in our uploading, so I am changing the name of the old files with the track of changes in order to prevent that problem.
In any case, it become clear a new revision was required as besides the mistakes identified by the reviewer(s) we identified an equal or larger number of mistakes. Probably is not a perfect product now, but we believe is much more improved now.
I hope you considered the manuscript is acceptable in the present shape.
Sincerely yours,
Juan Pablo and Philipp
This manuscript is a resubmission of an earlier submission. The following is a list of the peer review reports and author responses from that submission.
Round 1
Reviewer 1 Report
Dear authors, thank you for potentially very interesting article. Unfortunately there are number of flaws which seriously lower its scientific quality. Please refer to the attached file for details. The most important aspects which needs to be improved are: missing methods and any quantitative results which would support your conclusions. In many parts you present rather your subjective observations and suggestions than evidences. The last part where you describe remediation and prevention is entirely built on your feelings and without adding rigorous methods to examine these questions should be removed from the article.

Reviewer 2 Report
Overall and Major Comments
This paper analyzes a flood after a landslide as an example to illustrate the importance of new concept “forensic geology” and emphasizes that multidisciplinary research should be carried out as soon as possible after the occurrence of geological disasters. Main content of the paper focus on reconstructions of the two blocking events, results shows difference compared with published papers. Furthermore, authors emphasized importance of the “forensic geology”, which could support decision-maker of hazard putting forward better plan to deal with natural hazard. There are many weaknesses in the paper.
- The proof of “forensic geology” supporting better decision is weak in the paper, the discussion is based on many assumptions that without quantitative analysis. The discussion parts are weak. In my view, the more accurate reconstruction of these events is innovative, it seems unnecessary to discuss about decision making. Authors should focus the proposed new concept.
- The manuscript supports for reconstruction of the two blocking events other than the new concept, authors should add more content to support the importance of the new concept or change the title.
- There are too much characters in the abstract.
- In section 1.1, the title is the concept of “forensic geology”, but it is not explained in detail in text. And most of the subsequent content is irrelevant, this section needs to be rewritten. What amazed me is it has very limited references, and part of the content belongs to the subsequent part. For the paper, limited references were found.
- After reading the paper, the explanation of the character of the new concept is weak, I can’t distinguish between “forensic geology” and “detailed filed observation”.
- The dam breach happened in very short-time, it seems has limited time to do detailed field investigation before break. Author should add more text to explain how to use detailed investigation to support better decision.
- Please expound the mechanism of “molards” in the paper which is helpful for readers to understand it.
- Almost of all references are wrong, due to two numbers.
- In the conclusion section, it is mentioned that peer reviewed scientific journals have put forward a lot of views, please add relevant peer-reviewed citations The article does not give the machine learning model modeling process, this modeling process represents your understanding of this relationship.
- The description of field investigation is too much, author should weaken it and reinforce other part.
- The quality of English needs improved by native speaker.
Regarding above comments, a major revision should be requested. Many mistakes of writing and drawing were observed in the paper, authors should rewrite and redraw it carefully. Below, I present some of the specific comments.
L77: “do” should change into “did”.
L86: Where is the end of this sentence?
L87: Please redraw figure 2, the format and style is not in high quality.
L91: What is “white arrow”? In figure 2?
L93: I didn’t see 8 meters drop.
L95: What is “Grey arrow”? I think all the contents between L89-L98 belongs to figure 3 but the authors didn’t refer, which makes reader confusion.
L112-121: In this part, innovations of the paper is necessary to be described.
L128: What’s the meaning of “which later collapsed giving way to the jökulhlaup”. In the first paragraph of section 1.2, the authors could provide a short introduction of the events.
L144: In the unit, the “3” should be in top right of “m”.
L145: The unit of degree has no underline.
L145: The “reasonable stability” needs model to calculate.
L171: The plotting scale of 100 and 200 meter seems not match with each other.
L179: What is “Linear erosion”?
L249: Fig.5C should represent in detail, like showing the “diamictitic matrix” below.
L258: Figure6s’ location of “A” to “D” is different from figure 5.
L322: “mas” is wrong
L346: You said “(hatched line) suggesting the existence of internal lobes”, but how to prove it? If it was lobe, after dam breach, it should keep part of the lobe but it is plane in the figure. The slope of developing breach is around repose angle, it’s hard for me to image it could be plane after breach.
L351: Please mark the length in figure.
L344: The subtitle of figure b has three figures.
L365: Needs space between “wascarved”.
L703: Why it is 68m.
L704: Wrong word: “addiitonal”.
L875: Two full stops in the sentence.

Reviewer 3 Report
Several comments for the manuscript are as follows:
1. Title
• To deciphering → to decipher
• The title is on landslide dam collapse and outburst flood, yet the manuscript focusing on the formation.
2. Abstract
• L13 ‘as soon as possible after their occurrence’: this past is not connected with the ‘which is the multidisciplinary study of geohazards’ or the ‘demonstrate the importance of forensic geology’.
• L17-20: this part is focusing on what previous study are lacking, rather than focusing on the soundness of this study.
3. The manuscript
• L42-53: first paragraph focused on the definition of the word ‘forensic’ rather than explaining what is the importance of forensic (forensic geology) itself.
• The caption for the figures are nonexistent. Is it the first sentence of the following paragraph? Or is it the whole paragraph?
• L90-98: Explanation for Figure 8 suddenly appears here. Yet the explanation is disconnected with the figure. No white, black, or grey arrows in Figure 8.
• The authors often use ‘c.’ in front of a number, does it means ‘circa’? Circa is commonly used in front of a particular year. Please confirm this.
• Please check for typos. For example: L365 ‘wascarved’, L144 ‘2g/cm3’.
• L765-771, L799-802: are there any references for these?
• This manuscript only listed 10 references, is it due to lack of study of the same field? In that case, it can be one of the strength of this manuscript and can be emphasized in the introduction.